# Adipose-Derived Stem Cells (ADSCs) Supplemented with Hepatocyte Growth Factor (HGF) Attenuate Hepatic Stellate Cell Activation and Liver Fibrosis by Inhibiting the TGF-β/Smad Signaling Pathway in Chemical-Induced Liver Fibrosis Associated with Diabetes

**DOI:** 10.3390/cells11213338

**Published:** 2022-10-22

**Authors:** Sami Gharbia, Simona-Rebeca Nazarie, Sorina Dinescu, Cornel Balta, Hildegard Herman, Victor Eduard Peteu, Mihaela Gherghiceanu, Anca Hermenean, Marieta Costache

**Affiliations:** 1Department of Biochemistry and Molecular Biology, University of Bucharest, 050663 Bucharest, Romania; 2“Aurel Ardelean” Institute of Life Sciences, “Vasile Goldis” Western University of Arad, 310025 Arad, Romania; 3The Research Institute of the University of Bucharest (ICUB), University of Bucharest, 050663 Bucharest, Romania; 4Victor Babes National Institute of Pathology, 050096 Bucharest, Romania; 5Department of Cell Biology, Faculty of Medicine, “Carol Davila” University of Medicine and Pharmacy, 050474 Bucharest, Romania; 6Department of Histology, Faculty of Medicine, Vasile Goldis Western University of Arad, 310414 Arad, Romania

**Keywords:** liver fibrosis, diabetes mellitus, adipose-derived stem cells, hepatocyte growth factor, hepatic stellate cells, TGF-β/Smad pathway

## Abstract

Liver fibrosis can develop on the background of hyperglycemia in diabetes mellitus. However, xenobiotic-related factors may accelerate diabetes-associated liver fibrosis. In this study, we aimed to assess the antfibrotic effect of ADSC and HGF therapy and to establish the cellular and molecular mechanisms through in vitro and in vivo experiments. In vitro, TGF-β1-activated hepatic stellate cells (HSCs) were cocultured with ADSCs or HGF, and the expression of several fibrosis markers was investigated. The antifibrotic effect of the ADSCs, HGF, and ADSCs supplemented with HGF was further assessed in vivo on diabetic mice with liver fibrosis experimentally induced. In vitro results showed the inhibition of HSC proliferation and decrease in fibrogenesis markers. Coadministration of ADSCs and HGF on diabetic mice with liver fibrosis enhanced antifibrotic effects confirmed by the downregulation of Col I, α-SMA, TGF-β1, and Smad2, while Smad7 was upregulated. Moreover, stem cell therapy supplemented with HGF considerably attenuated inflammation and microvesicular steatosis, decreased collagen deposits, and alleviated liver fibrosis. In conclusion, the HGF-based ADSC therapy might be of interest for the treatment of liver fibrosis in diabetic patients, consecutive aggression exerts by different environmental factors.

## 1. Introduction

Diabetes mellitus (DM) is a chronic metabolic disorder that affects an increasing number of people globally and is characterized by elevated blood glucose (hyperglycemia) and elevated blood insulin (hyperinsulinemia) [1,2]. In many cases, DM is associated with long-term damage and dysfunction of other organs as well, such as chronic liver disease [3]. Liver fibrosis can develop in the background of hyperglycemia in DM and progress to more advanced stages of liver injury, such as cirrhosis and hepatocellular carcinoma, which could further lead to the patient’s death [4,5]. In liver fibrosis, the cellular key player is represented by hepatic stellate cells (HSCs), which are activated in the context of inflammation or liver injury and produce components of the extracellular matrix (ECM), which lead to the accumulation of scar tissue [6]. They follow two activation phases, the initiation step triggered by paracrine stimulation of other liver cells and the perpetuation phase, which maintains their activated status [7]. They transdifferentiate from a quiescent state to a proliferative myofibroblast-like phenotype, express alpha-smooth muscle actin (α-SMA), and produce collagen type I (Col I), the main component of accumulating ECM [8]. ECM changes its composition in liver fibrosis, with many types of collagens (I, III, IV), fibronectin, vimentin, and so on. Moreover, the excessive accumulation of ECM in liver fibrosis is driven by reduced activity of metalloproteinases (MMPs) and increased expression of tissue inhibitors of MMPs (TIMPs) [9]. One of the factors that stimulate HSC activation is transforming growth factor β (TGF-β). It can further activate SMAD molecules to phosphorylate and form a complex between Smad 2 and 3, which translocates to the nucleus and induces the transcription of other fibrotic genes [10].

One important connection on how DM promotes liver fibrosis is given by the profibrogenic influence of glucose and insulin on HSC activation [11]. As DM is frequently associated with systemic inflammation, this condition could stimulate the progression of liver fibrosis [12,13]. Furthermore, liver fibrosis is promoted in DM patients by other mechanisms as well: hepatocyte apoptosis induced by dysregulation of the insulin receptor pathway [14,15], as well as angiogenesis associated with DM, also promotes fibrosis [16,17].

The use of some type of treatments for DM based on hypoglycemic agents is restricted for patients with chronic liver diseases because of the potential hepatotoxic effect [3]. Stem-cell-based therapies have gained increasing interest for many diseases, mostly based on their versatility and multiple applications. Not only the rich secretome of stem cells (SCs) has therapeutic potential, but the SCs themselves can help to stimulate regeneration by hepatogenic differentiation after transplantation [18,19,20]. Moreover, choosing the patient’s own mesenchymal stem cells (MSCs), such as adipose-derived stem cells (ADSCs), which are so easily obtainable and have low immunogenicity, eliminates the rejection risk, in case donor stem cells are needed [21,22]. MSCs’ secretome contains soluble proteins, free nucleic acids, and extracellular vesicles (EVs) that vary depending on the MSCs’ organ source [23,24]. The many positive effects proven so far for MSCs’ secretome, such as immunomodulatory, anti-inflammatory activity, antiapoptotic activity, regulation of angiogenesis, and wound healing [23], recommend its use as possible treatment for liver fibrosis associated with DM. In addition, other reports suggested that ADSC therapy ameliorated hyperglycemia and insulin resistance in the animal model of type 2 diabetes [2,25,26].

However, the environmental and xenobiotic-related factors may accelerate the hepatic pathogenesis and aggravate diabetes-associated liver disease [27,28,29]. Some of these xenobiotics may result in harmful effects on the cellular components of the liver, leading to increased wound healing response and fibrogenesis activation, which in the long-term can lead to liver fibrosis and cirrhosis [27]. Therefore, in the context of the evolution of diabetes and the appearance of subsequent complications, chemical-induced liver injury can evolve much faster towards fibrosis, respectively cirrhosis, and therefore, effective therapies to prevent harmful effects are needed.

To date, it is known that ADSCs can secrete many growth factors, which contribute to tissue remodeling mainly through paracrine mechanisms [30,31]. Of these, the hepatocyte growth factor (HGF) is a tissue growth factor for the promotion of hepatocyte regeneration and also plays an important role in the prevention of tissue fibrosis and apoptosis [32,33,34]. Moreover, it was shown that ADSCs that overexpress HGF exerted a better therapeutic effect in an acute myocardial infarction rat model [35]. On the contrary, when HGF expression is downregulated, the efficacy of ADSCs in the repair of ischemic tissue is altered [36]. Similarly, HGF, which physiologically plays a hepatotropic role for liver regeneration, seems to exert protective effects against different types of liver injuries and exhibits antifibrotic action for the liver in rodent models [37,38] or induced recovery from the alcohol-induced fatty liver in rats [39].

Despite the fact that there are promising results regarding the regenerative properties of ADSC or HGF therapy, there are no studies so far that highlight the ability of ADSC and HGF cotreatment to reduce liver fibrosis in the background of pre-existing diabetes.

In this study, we aimed to assess the antifibrotic effect of ADSC and HGF therapy and to establish the cellular and molecular mechanisms through in vitro and in vivo experiments. In vitro, HSCs were cocultured with ADSCs or HGF in order to evaluate the potential antifibrotic effect of the ADSCs’ secretome and HGF, and the expression of several fibrosis markers was investigated at gene and protein levels. The antifibrotic effect of the ADSCs, HGF, and ADSCs supplemented with HGF was further assessed in vivo on diabetic mice with liver fibrosis chemically induced by the analysis of histopathological and electron microscopy findings, IHC/IF staining, and gene expression of the main pathways responsible for liver fibrogenesis.

## 2. Materials and Methods

### 2.1. In Vitro Experiments

#### 2.1.1. Cell Cultures and In Vitro Coculture Experimental Model

Cell cultures of human hepatic stellate cells from the LX2 cell line were obtained (kindly provided by Dr. Pau Sancho-Bru with permission from Dr. Scott Friedman, Icahn School of Medicine at Mount Sinai, New York, NY, USA) [40]. HSCs were cultured in Dulbecco’s Modified Eagle Medium (DMEM, Sigma-Aldrich, St. Louis, MI, USA) supplemented with 2% fetal bovine serum (FBS, Gibco) and 1% antibiotic antimycotic solution (ABAM, Sigma-Aldrich). HSCs were activated (aHSCs) with 10 ng/mL TGF-β1 for 24 h (Cell Signaling, Danvers, MA, USA) [41,42]. Treatment with 50 ng/mL HGF (Thermo Fischer Scientific, Waltham, MA, USA, R7788115) was conducted after HSC activation for 24 h [43]. For the coculture conditions, human adipose-derived stem cells (ADSCs, Thermo Fischer Scientific, R7788115) were purchased and propagated in culture in DMEM supplemented with 10% FBS and 1% ABAM. Cell cultures were maintained at 37 °C, 5% CO_2_ with humidity. For the in vitro experimental model, 24-well plates (for immunofluorescence staining) at 1.5 × 10^4^ cells/cm^2^ and 6-well plates (for gene expression evaluation) at 1 × 10^4^ cells/cm^2^ were seeded with HSCs and TGF-β1-activated. After activation, cell culture insert plates (VWR International, Radnor, PA, USA) were added with cultured ADSCs at 1 × 10^4^ cells/cm^2^ for 48 h. Five conditions were established for further IF and qPCR analysis: LX2 (control HSCs from LX2 cell line), activated LX2 (HSCs from LX2 cell line treated with 10 ng/mL TGF-β1), activated LX2 + ADSC (activated LX2 cocultured with ADSC), activated LX2 + HGF (activated LX2 treated with 50 ng/mL HGF), activated LX2 + ADSC + HGF (activated LX2 cocultured with ADSC, treated with 50 ng/mL HGF).

#### 2.1.2. Immunofluorescence Staining (IF) of Fibrosis Markers

HSCs cocultured with ADSC were fixed with 4% paraformaldehyde solution for 1 h, and permeabilization was performed with 2% bovine serum albumin (BSA, Sigma-Aldrich) solution with 0.1% Triton X-100 for 20 min. This was followed by overnight incubation at 4 °C with specific antibodies: anti-α-SMA (sc-53015, Santa Cruz Biotechnology, Dallas, TX, USA), anti-SMAD2/3 (5678S, Cell Signaling), anti-COL1A1 (72026, Cell Signaling), and antifibronectin (sc-69681, Santa Cruz Biotechnology). Samples were incubated with specific anti-rabbit/anti-mouse secondary antibodies (Thermo Fischer Scientific) conjugated with AF 546 and AF 488, respectively, and further were stained with Hoechst 33342 (Sigma-Aldrich) for 5 min to visualize the nuclei in blue. Samples were visualized by fluorescence microscopy (IX-73 Olympus).

#### 2.1.3. Gene Expression Evaluation by Real-Time PCR (qPCR)

Total RNA was extracted from the HSCs cocultured with ADSCs with TRIzol Reagent (Thermo Scientific, Waltham, MA, USA). The concentration and purity of extracted RNA samples were determined on a NanoDrop 8000 spectrophotometer (Thermo Fisher Scientific, Waltham, MA, USA). Total RNA was reverse-transcribed, and cDNA was obtained using the iScript cDNA synthesis kit (Bio-Rad) on a Veriti 96-Well Thermal Cycler (Applied Biosystems). Real-time PCR was performed following the kit’s instructions (SYBR Select Master Mix, Thermo Scientific, Waltham, MA, USA), and the ViiA 7 Real-Time PCR System (Thermo Scientific, Waltham, MA, USA) was used. Samples were tested in triplicate, and the housekeeping gene used was glyceraldehyde 3-phosphate dehydrogenase (*gapdh*). Sequences of the primers were the following: *α-sma* F 5′ TTCGCATCAAGGCCCAAGAA 3′, *α-sma* R 5′ GTCCCGGGGATAGGCAAAG 3′, *col1a1* F 5′ TCCTGGTCCTGCTGGCAAAGAA 3′, *col1a1* R 5′ CACGCTGTCCAGCAATACCTTGA 3′, *fibronectin* F 5′ CCATCGCAAACCGCTGCCAT 3′, *fibronectin* R 5′ AACACTTCTCAGCTATGGGCTT 3′, *gapdh* F 5′ GAGTCAACGGGGTCGT 3′, and *gapdh* R 5′ TTGATTTTGGATCTCG 3′. Results were statistically analyzed with GraphPad Prism 6.0, one-way ANOVA, and Bonferroni correction; *p*-values < 0.05 were considered statistically significant in accordance with the 95% confidence interval established for the analysis.

### 2.2. In Vivo Experiments

#### 2.2.1. Animals and In Vivo Experimental Design

Prior to the experiment, adult CD1 mice (weighing 28–30 g) from the Animal Facility of the Vasile Goldiș Western University of Arad were housed at 20–25 °C under a standard 12/12 light–dark cycle with 60% relative humidity. The mice had *ad libitum* access to food and autoclaved water. All animal procedures were approved by the Animal Ethics Committee of the Vasile Goldiș Western University of Arad (155/02.10.2019). For the in vivo study, 10 mice were included randomly in each experimental group.

After overnight fasting, mice were intraperitoneally (i.p.) injected with a single dose of streptozotocin (STZ; 102 mg/kg b.w.), freshly dissolved in citrate buffer (50 mM, pH 4.5) to induce diabetes [44]. Blood glucose levels were measured after 4 h of fasting by using a one-touch glucometer. After STZ administration, mice with a fasting blood glucose level higher than 200 mg/dL for 2 consecutive weeks were considered diabetic mice. Confirmed diabetic mice were further treated i.p. with CCl_4_ dissolved in olive oil (20% *v*/*v*, 2 mL/kg), twice a week for 7 weeks, and euthanatized 72 hours after the last injection for confirmation of liver fibrosis (liver fibrosis diabetes mice: LF), according to our previous work [45]. Spontaneous resolution of hepatic fibrosis was investigated in CCl_4_-treated diabetic animals after 2 weeks of self-recovery (liver fibrosis control diabetes mice: LFC). The treatments started on week 7, at the end of the CCl_4_ administration, and received: ADSCs cells (1.0 × 10^6^ cells/100 μL PBS; Thermo Fischer Scientific, R7788115) by i.v. injection on the tail vein, on week 7 (ADSCs diabetes mice: ADSC); hepatocyte growth factor (150 μg/kg) by i.v. injection on the tail vein, on week 7 (HGF diabetes mice: HGF); and ADSCs plus HGF in the same doses as previous groups administrated by i.v. injection on the tail vein, on week 7 (ADSCs and HGF diabetes mice: ADSC + HGF) (Figure 1). The control group received saline solution. After blood collection, the animals were euthanized by administering an overdose of ketamine. Our preliminary studies established the HGF dose and according to a previous data field [39,46].

#### 2.2.2. Histology and Immunohistochemistry

Liver samples were fixed in a 4% formaldehyde solution in PBS and embedded in paraffin. Sections were stained with Gomori’s trichrome stain kit (38016SS1, Leica, USA). Histological analysis was performed using an Olympus BX43 microscope.

The 5 μm sections were deparaffinized, rehydrated, and incubated overnight at 4 °C with the primary antibodies TGF-β1(sc-146). The system detection Novolink Max Polymer (Leica Biosystems, Wetzlar, Germany) and 3,30-diaminobenzidine (DAB) as a chromogenic substrate were used for immunodetection. The nuclei were counterstained with hematoxylin. Images were assessed by light microscopy (Olympus BX43, Hamburg, Germany).

#### 2.2.3. Immunofluorescence

For immunofluorescence, the deparaffinized, rehydrated liver sections were incubated overnight in epitope retrieval solution (Leica Biosystems Inc., Buffalo Grove, IL, USA) at 60 °C, followed by blocking with PBS containing 2% bovine serum albumin (BSA) (ABIN934476, Antibodies-online) and 0.1% Triton X-100 (X-100, Sigma-Aldrich, St Louis, MI, USA) at room temperature for 40 min. The tissue sections were incubated with rabbit anti-Smad 2/3 (sc-8332, 1:300 dilution; Santa-Cruz, CA, USA) and rabbit anti-α-SMA (ab32575; 1:500 dilution; Abcam, Cambridge, UK) primary antibodies for 2 h at room temperature. The primary antibodies were diluted in primary antibody diluting buffer (Bio-Optica, Milano, Italy). The slides were washed in PBS three times and incubated with the Alexa Fluor 594 labeled goat anti-rabbit IgG secondary antibody (A 11037; Thermo Fisher Scientific Inc., Rockford, USA), diluted to 1:400 in PBS, for 30 min at room temperature in the dark. After three more washing steps with PBS, the sections were counterstained with 4′,6-diamidino-2-phenylindole (DAPI) for 5 min, washed by PBS, and mounted with CC/Mount aqueous mounting medium (Sigma-Aldrich, St Louis, MI, USA). Images were captured using a Leica TCS SP8 laser scanning confocal microscope.

#### 2.2.4. Transmission Electron Microscopy

Liver samples were prefixed in a 2.7% glutaraldehyde solution in a 0.1 M phosphate buffer, washed in a 0.15 M phosphate buffer (pH 7.2), postfixed in a 2% osmic acid solution in a 0.15 M phosphate buffer, dehydrated in acetone, and then embedded in the epoxy embedding resin Epon 812. Further, 70 nm sections were double-contrasted with uranyl acetate and lead citrate and analyzed with a TEM microscope (Morgagni 268, FEI, Eindhoven, Netherlands). Data acquisition was performed with a MegaView III CCD using the iTEM SIS software (Olympus Soft Imaging Software, Munster, Germany).

#### 2.2.5. Quantitative Real-Time PCR Analysis

Real-time quantitative polymerase chain reaction (qPCR) was applied to assess mRNA expression of TGF-β1, Smad 2/3, Smad 7, Col I, and α-SMA. Total RNA was extracted using the SV Total RNA isolation kit (Promega), and the quantity and quality were assessed using a spectrophotometer (NanoDrop One, Thermo Scientific, Waltham, MA, USA), then the reverse transcription performed using the First Strand cDNA Synthesis Kit (Thermo Scientific, Waltham, MA, USA). RT-PCR was performed using the Maxima SYBR Green/ROX qPCR Master Mix (Life Technologies, Carlsbad, CA, USA) with an Mx3000PTM RT-PCR system. All samples were run in triplicate. The primers are shown in Table 1. Glyceraldehyde 3-phosphate dehydrogenase (GAPDH) was used as a reference gene and was assessed under the same experimental protocol. Relative expression changes were determined using the 2 ∆∆ C(T) method [47].

#### 2.2.6. Statistical Analysis

Data analyses were performed using GraphPad Prism 9.4.0. Results are expressed as mean ± SD. Statistical analyses were performed by employing analysis of variance (one-way ANOVA) with the Bonferroni correction. A *p*-value of <0.05 was considered significant.

The quantification of α-SMA and COL I levels was made with the ImageJ software, and plots were obtained with GraphPad Prism version 6 for Windows.

## 3. Results

### 3.1. Immunofluorescence Staining (IF) of Fibrosis Markers

The results of immunofluorescence staining of fibrosis markers (α-SMA, COL I, Smad 2/3) are presented in Figure 2, Figure 3 and Figure 4. As shown in Figure 2, when activated with TGF-β1, HSCs expressed high levels of α-SMA compared with control. When exposed to ADSCs in coculture, α-SMA expression in HSCs was significantly reduced, suggesting that the secretome of ADSCs contained important antifibrosis factors that acted upon aHSCs. One of these factors could be HGF, as it reduced the expression of α-SMA in HSCs after treatment. Moreover, α-SMA expression was also reduced in HSCs cocultured with ADSCs and further treated with HGF.

Another important HSC activation marker is represented by COL I, and its expression in HSCs is shown in Figure 3. COL I expression was induced in aHSCs by TGF-β1 treatment. In HSCs cocultured with ADSCs, COL I expression was reduced, suggesting the effect of ADSCs’ secretome on aHSCs reversion. After HGF treatment, the expression of COL I was significantly reduced in a similar manner to the ADSC coculture effect, which could indicate that the effect of ADSCs’ secretome could be associated with the presence of HGF. In HSCs exposed to ADSCs and HGF, the expression of COL I was significantly reduced, indicating that HGF and ADSCs’ secretome potentiate each other and enhance their effect. Moreover, the effect of the ADSCs’ secretome on HSCs was investigated by the evaluation of the cellular localization of the Smad 2/3 complex (Figure 4). Upon activation of the TGF-β/Smad signaling pathway, the Smad 2/3 complex is formed and transfers to the nuclei. Upon TGF-β1 exposure, the Smad 2/3 complex translocates to the nuclei, as shown in Figure 4. However, in HSCs exposed to ADSCs, the Smad 2/3 complex is mostly located in the whole cell, suggesting the inhibition of the TGF-β/Smad pathway. Treatment with HGF also inhibited the translocation of the Smad 2/3 complex in the nucleus. In addition, HSCs cocultured with ADSCs and treated with HGF presented a similar response, inhibiting the transfer of the Smad 2/3 complex in the nucleus.

### 3.2. Gene Expression Evaluation by Real-Time PCR (qPCR)

The immunofluorescence staining results were confirmed by gene expression results. Gene expression evaluation by qPCR was conducted for α-SMA and COL1A1 (Figure 5). The elative gene expression of α-SMA was statistically significantly increased ~5x in HSCs after TGF-β1 treatment (Figure 5a). In HSCs cocultured with ADSCs, α-SMA expression was statistically significantly (*p* < 0.001) reduced to levels similar to control. Additionally, the expression of α-SMA was statistically significantly (*p* < 0.001) reduced after treatment with HGF as well as when HSCs were both cocultured with ADSCs and treated with HGF. This confirms the IF results and suggests the role of ADSCs’ secretome and HGF in stimulating aHSCs’ reversion. In addition, the expression of *col1a1* was also assessed, and the COL1A1 results of IF were confirmed at gene levels as well. The expression of *col1a1* was upregulated after TGF-β1 treatment, which was later statistically significantly downregulated (*p* < 0.001) upon coculture with ADSCs, but also in the presence of HGF, suggesting the effect of ADSCs’ secretome and HGF on the secretion of extracellular matrix components.

### 3.3. ADSC and HGF Cotreatment Inhibit Activation of Hepatic Stellate Cells (HSCs) in Fibrotic Livers of Diabetic Mice

The expression of α-SMA highlights the activated HSCs and is considered to be one of the important markers of hepatic fibrosis. The RT-PCR analysis showed significant upregulation of the α-SMA gene expression for the LF group (*p* < 0.001). The ADSC, HGF, and ADSC + HGF groups presented significantly decreased levels of gene expression compared with the LF group (Figure 6A). The best results were obtained for the cotreatment of ADSCs and HGF, where a significant decrease was obtained compared with both the LF and LFC groups. The immunofluorescence analysis showed immunopositivity for the slides of fibrotic livers and reduced almost to the level of control after the treatment with stem cells and HGF, while de novo recovery of the liver without any treatment (LFC) was close to the LF (Figure 6B).

### 3.4. ADSCs and HGF Suppress the Production of the Collagen in a Liver Fibrosis Model of Diabetic Mice

Collagen proliferation was evidenced using Gomori’s trichrome stain kit (Figure 6A). The control liver had a normal lobular structure without any proliferation of collagen, while diabetic mice with liver fibrosis showed significant collagen deposition and formation of pseudo-lobules. The extent of fibrotic change was still noticed after 2 weeks of toxic cessation. Treatment with ADSCs, HGF, and ADSCs + HGF reduced the thickness of fibrous septa, and for cotreatment, the best results were obtained. These results were confirmed by collagen 1 downregulation (Figure 7B).

### 3.5. ADSC and HGF Cotreatment Downregulate TGF- β1/Smad Signaling in Fibrotic Livers of Diabetic Mice

TGF-β1 is considered an essential promoter of fibrogenesis and acts through Smad 2/3 phosphorylation, while Smad 7 is an inhibitor of this pathway. mRNA analysis showed an upregulation of the TGF-β1 gene expression and a strong immunopositivity for fibrotic fivers of the LF group (Figure 8). The TGF-β1 mRNA level was significantly reduced by ADSC, HGF, or ADSC + HGF treatments (*p* < 0.001). Similarly, the hepatic mRNA level and immunofluorescence of Smad2 were reduced in the same pattern. In contrast, ADSC, HGF, or ADSC + HGF treatments considerably increased the expression of the mRNA Smad7 compared with the fibrotic group.

### 3.6. ADSC and HGF Cotreatment Improve Liver Function and Morphology of Fibrotic Livers in Diabetic Mice

To evaluate the potential therapeutic ability of ADSCs and HGF to reverse liver fibrosis in a diabetic mouse model, we injected CCl_4_ i.p. for 7 consecutive weeks. The histological analysis of the controls showed a normal liver lobular architecture with central vein and radiating hepatic cords, without any proliferation of connective tissue (Figure 9(Aa)), which was confirmed by electron microscopy (Figure 9(Ca)). Liver specimens from the LF group showed severe changes in morphology, including microvesicular steatosis with the presence of clusters of foam cells. Parenchymal collagen deposition, formation of pseudolobules, and infiltration of inflammatory cells were noticed. Moreover, electron microscopy analysis of the fibrotic livers reveals the presence of activated hepatic stellate cells and massive accumulation of lipids into foam cells, smooth endoplasmic reticulum proliferation and bundle of collagen fibers proliferated into the parenchyma, space of Disse and between swollen profiles of sinusoidal endothelial cells, and glycogen depletion (Figure 9(Bb–d)). The structural and ultrastructural alterations were observed in the self-recovery group (LFC), similar to fibrotic livers (Figure 9(Ac,Be)). After treatment with ADSCs, HGF, and ADSCs + HGF, both the hepatic structure and ultrastructure were significantly restored, and the best improvement was obtained for the ADSCs + HGF group (Figure 9(Af,Bh)).

## 4. Discussion

HGF is known as a growth factor secreted by ADSCs [30,31], which contributes to liver recovery after injury [37,38]. Thus, we focused on demonstrating the antifibrotic effect of HGF-based ADSC therapy and establishing the cellular and molecular mechanism by using an in vitro model of activated HSCs, the key player of liver fibrogenesis, and an in vivo model of CCl_4_-induced liver fibrosis in diabetic mice.

In our study, we evaluated first the effect of ADSCs’ secretome and HGF in vitro on HSCs that were activated by TGF-β1 treatment. We showed that in HSCs cocultured with ADSCs or HGF, the expression levels of α-SMA and collagen type I were statistically significantly reduced compared with activated HSCs, at both the gene and protein levels. This suggests that both ADSCs’ secretome and HGF influence the behavior of HSCs and inhibit their activation. This was also confirmed in a study by Yu et al. in 2015 [48]. The authors cocultured rat HSCs with ADSCs or HGF and showed that HSC proliferation was inhibited and their apoptosis was promoted. Moreover, the expression of α-SMA and collagen type I was significantly reduced in HSCs cocultured with ADSCs or HGF.

One of the most important signaling pathways activated in liver fibrosis is TGF-β1/SMAD [49]. Once the pathway is activated, phosphorylated SMAD2 and SMAD3 molecules form a complex that translocates to the nucleus and induces the transcription of fibrosis markers. We investigated the cellular distribution of the SMAD 2/3 complex in HSCs before and after coculture with ADSCs. The complex was found in the nuclei after TGF-β1 activation in HSCs, but after coculture with ADSCs or HGF, it was localized in the whole cell, suggesting the effect of the ADSCs’ secretome to inhibit the TGF-β1/SMAD fibrogenic pathway. In a study by Liao et al. [2], rats induced with type II diabetes (T2D) and CCl_4_-induced liver fibrosis were further transplanted with ADSCs via tail vein injection. The expression of TGF-β1, SMAD3, and p-SMAD3 was investigated, and they found that after ADSC transplantation or HGF therapy, the phosphorylation of SMAD3 was inhibited, suggesting the effect of ADSCs and HGF to downregulate the TGF-β1/SMAD3 signaling pathway.

Following the promising results obtained in vitro, we investigated the therapeutic potential of ADSCs supplemented with HGF to alleviate or reverse liver fibrosis induced experimentally in diabetic mice.

The coadministration of ADSCs and HGF potentially reversed the fibrotic process by inhibiting collagen deposition and TGF-β1, Smad2, and α-SMA production, while Smad 7 (an inhibitor of the profibrotic signaling) was increased, which highlights the possible mainly molecular antifibrotic mechanism. Our results are in line with previous studies showing that inhibiting the transforming growth factor-β1-Smad signaling pathway may account for the antifibrosis effect of MSCs [50]. Complementarily, we highlighted the inhibitory effects on transdifferentiated hepatic stellate cells, as we observed by electron microscopy analysis of the micrographs from the diabetic mice with liver fibrosis treated with ADSCs and HGF in this experimental group. In this study, we also tested for the immunofluorescence expression of α-SMA and gene expression, which is associated with the activation of hepatic stellate cells. ADSCs supplemented with HGF therapy were able to significantly mitigate CCl_4_-induced liver fibrosis in diabetic mice by inhibiting the α-SMA and collagen-I expression, results that highlight the key role of HGF in liver recovery after injury.

After the cessation of the chronic damage, the liver fibrosis resolution’s mechanism involved gradual steps, from shifting the inflammatory macrophages that acquire a restorative phenotype to deactivation and elimination of myofibroblasts and, finally, the extracellular matrix degradation [51]. A key process in fibrosis regression represents the senescence, apoptosis, and inactivation of hepatic stellate cells (HSC), which are the main collagen-producing cells in the liver by transdifferentiation into myofibroblasts during fibrogenesis [52]. After only 2 weeks of treatment, we obtained a significant inhibition of the TGF-β/Smad pathway for the ADSC + HGF group compared with the fibrotic groups and the other individual treatments, which suggested that the cotreatment initiated the deactivation process of HSCs. However, there is no significant difference in gene expression for collagen between the groups treated with stem cells or HGF and, respectively, the cotreatments, and probably, a longer analysis interval would be necessary to allow the gradual activation of resolution mechanism steps and to be transposed in a significant difference in the collagen deposition.

To date, it is assumed that MSCs secrete HGF, endothelial growth factor (VEGF), epidermal growth factor (EGF), MMP family proteins, and other cytokines used further in tissue repair [32,33,34]; among them, HGF is deeply involved in liver regeneration after injury. HGF binds to its specific receptor c-Met (cellular mesenchymal–epithelial transition) and triggers the intracellular intrinsic kinase activity of c-Met, affecting cell proliferation, growth, and survival [53]. In this respect, HGF has important clinical significance to improve liver fibrosis, hepatocyte regeneration after inflammation, and liver regeneration after transplantation [53]. It is probable because HGF exerts biological activities in regulating lipid metabolism, as well as in stimulating hepatocyte proliferation through the c-Met/HGF receptor, as was previously demonstrated in acute alcoholic hepatitis [53]. Another hypothesis is that HGF-overexpressing ADSCs displayed a better antifibrosis therapeutic efficacy in chronic diabetes, because it plays a crucial role in the mobilization, migration, and homing of MSCs [54].

Hepatocyte growth factor (HGF) secreted by various stem cells plays a critical role in their antifibrotic effects. Umbilical cord mesenchymal stem cells promoted liver repair by secreting HGF in [55], while menstrual blood-derived mesenchymal stem cells suppressed activated hepatic stellate cells via the paracrine activation of HGF and other mediators in [56], and the endothelial progenitor cells’ transplantation induced beneficial effects in carbon tetrachloride-induced liver fibrosis by activated HGF-mediated hepatocyte proliferation in [57]. Moreover, ADSCs’ conditioned media inhibited the proliferation of fibroblasts derived from a human hypertrophic scar in a dose-dependent manner via HGF-like protein in [58], which supports our hypothesis that ADSCs and HGF have a positive feedback loop and further enhance their protective role in liver fibrosis.

## 5. Conclusions

ADSC therapy and HGF administration showed the potential to reduce the expression of fibrosis markers in HSCs and downregulate the TGF-β/Smad fibrogenic pathway. Moreover, the therapeutic effects of ADSCs were highlighted in vivo on liver fibrosis chemically induced in diabetic mice when supplemented with HGF. However, HGF administration could be much more convenient and economical when compared with the HGF-based ADSC therapy and might be of interest for the treatment of liver fibrosis in diabetic patients, consecutive aggression exerts by different environmental factors. Moreover, it can be extended to other complications of chronic diabetes, such as diabetic nephropathy, diabetic retinopathy, or cardiac fibrosis.

## Figures and Tables

**Figure 1 cells-11-03338-f001:**
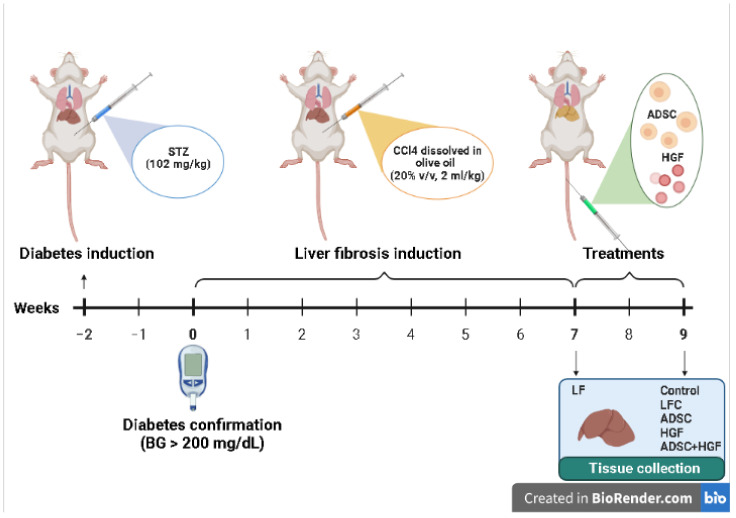
The experimental design of the in vivo experiment.

**Figure 2 cells-11-03338-f002:**
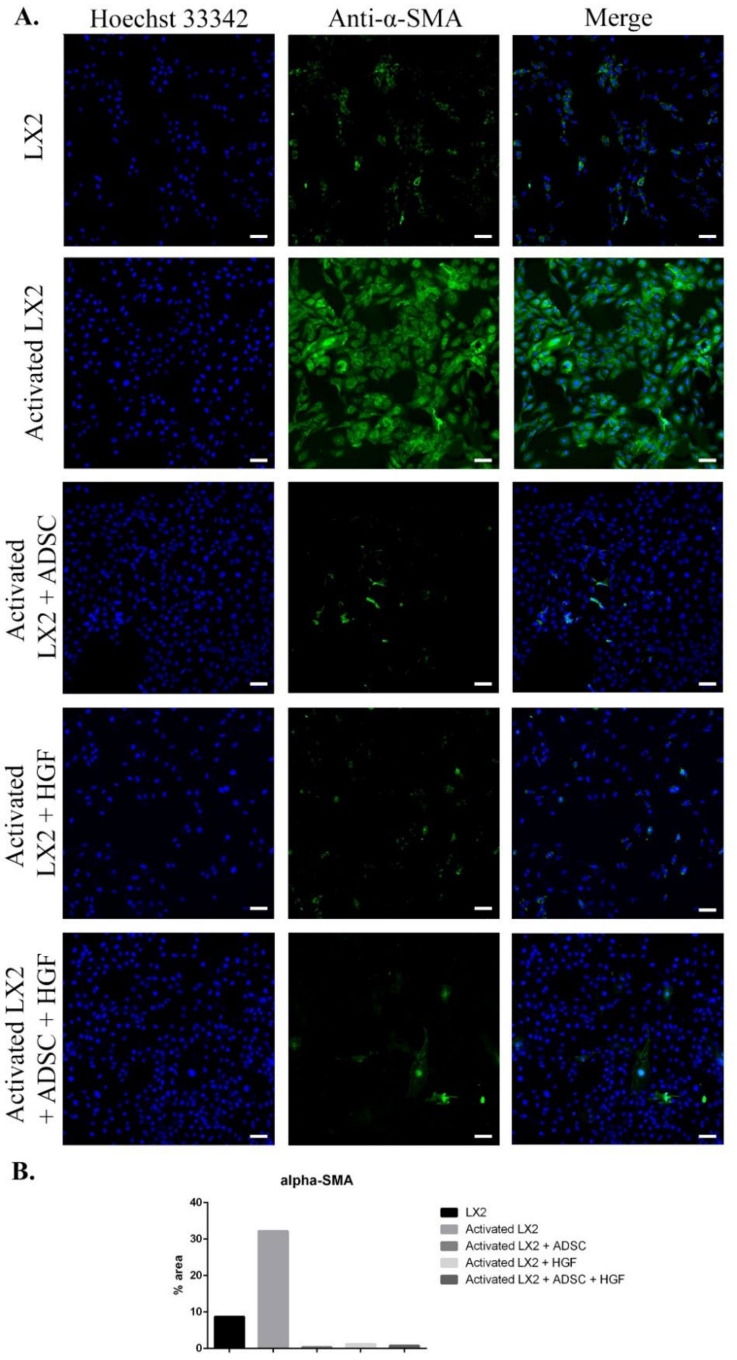
Evaluation of α-SMA expression in the following conditions: LX2, activated LX2 (induced with TGF-β1), activated LX2 + ADSC (activated LX2 cocultured with ADSCs), activated LX2 + HGF (activated LX2 treated with HGF), activated LX2 + ADSC + HGF (activated LX2 co-cultured with ADSC, treated with HGF). (**A**) Immunofluorescence staining of α-SMA (antibody conjugated with AF-488, green), nuclei of the cells stained with Hoechst 33342 (blue); (**B**) quantification of the immunofluorescence staining of α-SMA; scale bar: 100 µm.

**Figure 3 cells-11-03338-f003:**
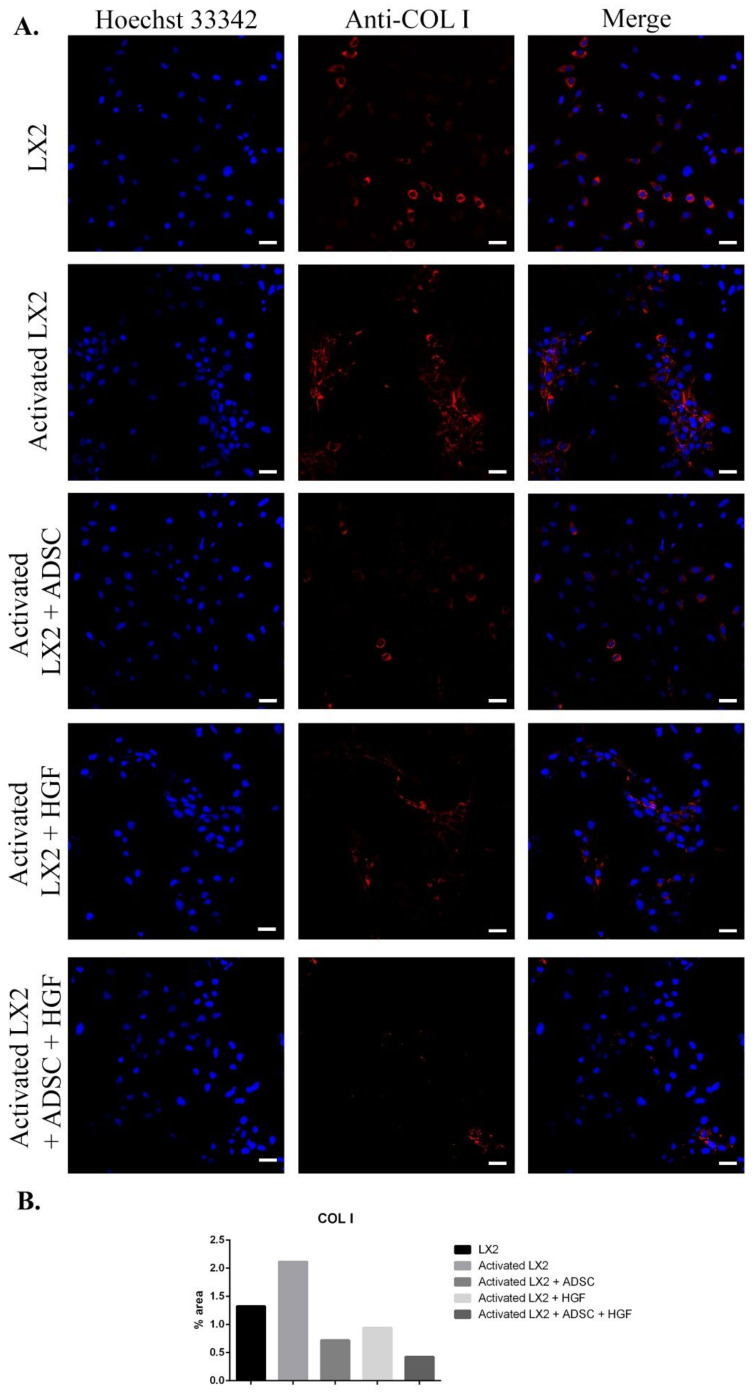
Evaluation of collagen type I (COL I) expression in the following conditions: LX2, activated LX2 (induced with TGF-β1), activated LX2 + ADSC (activated LX2 cocultured with ADSCs), activated LX2 + HGF (activated LX2 treated with HGF), activated LX2 + ADSC + HGF (activated LX2 cocultured with ADSC, treated with HGF). (**A**). Immunofluorescence staining of COL I (antibody conjugated with AF-546, red), nuclei of the cells stained with Hoechst 33342 (blue); (**B**) quantification of the immunofluorescence staining of COL I; scale bar: 50 µm.

**Figure 4 cells-11-03338-f004:**
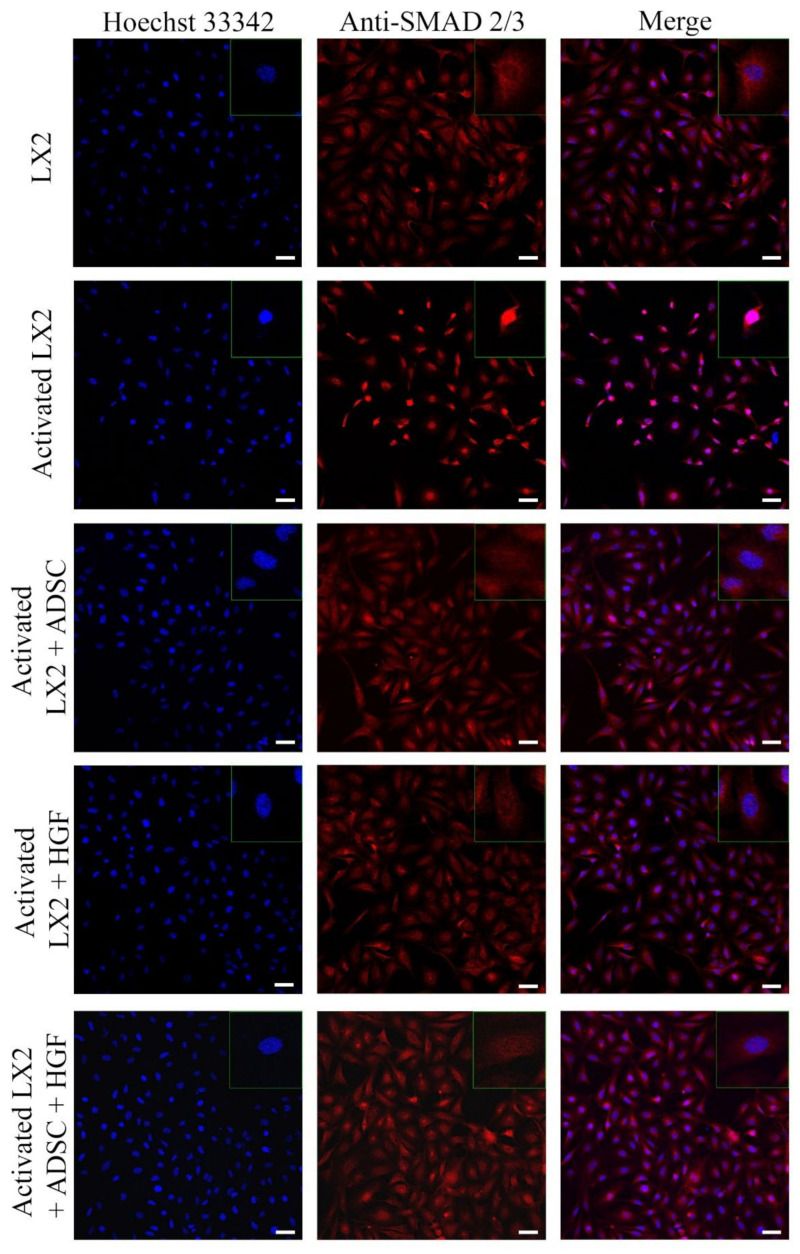
Immunofluorescence staining of SMAD 2/3 (antibody conjugated with AF-546, green), nuclei of the cells stained with Hoechst 33342 (blue), in the following conditions: LX2, activated LX2 (induced with TGF-β1), activated LX2 + ADSC (activated LX2 cocultured with ADSCs), activated LX2 + HGF (activated LX2 treated with HGF), activated LX2 + ADSC + HGF (activated LX2 cocultured with ADSC, treated with HGF); scale bar: 50 µm.

**Figure 5 cells-11-03338-f005:**
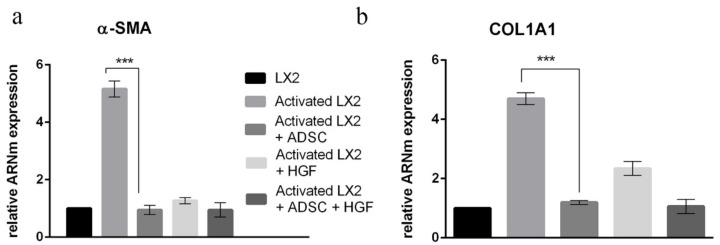
Evaluation of α-SMA (**a**) and COL1A1 (**b**) expression in HSCs by qPCR. Statistical significance: *** *p* < 0.001.

**Figure 6 cells-11-03338-f006:**
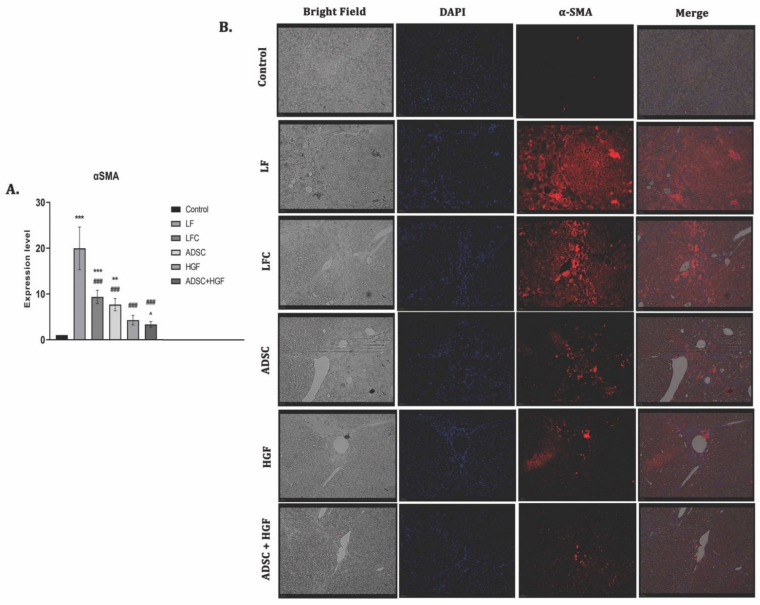
ADSCs and HGF cotreatment on α-SMA induced hepatic stellate cell (HSC) activation (**A**). RT-PCR analysis of the α-SMA gene level. Legend: Control, LF—liver fibrosis in diabetic mice; LFC— self-recovery of liver fibrosis (positive control) in diabetic mice; ADSC—ADSC treatment of liver fibrosis in diabetic mice; HGF—HGF treatment of liver fibrosis in diabetic mice; ADSC + HGF—ADSC and HGF cotreatment of liver fibrosis in diabetic mice; *** *p* < 0.001 compared with control; ** *p* < 0.01 compared with control; ### *p* < 0.001 compared with the LF-DIA group; ^ *p* < 0.05 compared with the LFC group (**B**). Immunofluorescence expression of α-SMA in experimental livers. Control, LF–liver fibrosis in diabetic mice; LFC–self-recovery of liver fibrosis (positive control) in diabetic mice; ADSC—ADSC treatment of liver fibrosis in diabetic mice; HGF—HGF treatment of liver fibrosis in diabetic mice; ADSC + HGF—ADSC and HGF cotreatment of liver fibrosis in diabetic mice; scale bar: 50 µm.

**Figure 7 cells-11-03338-f007:**
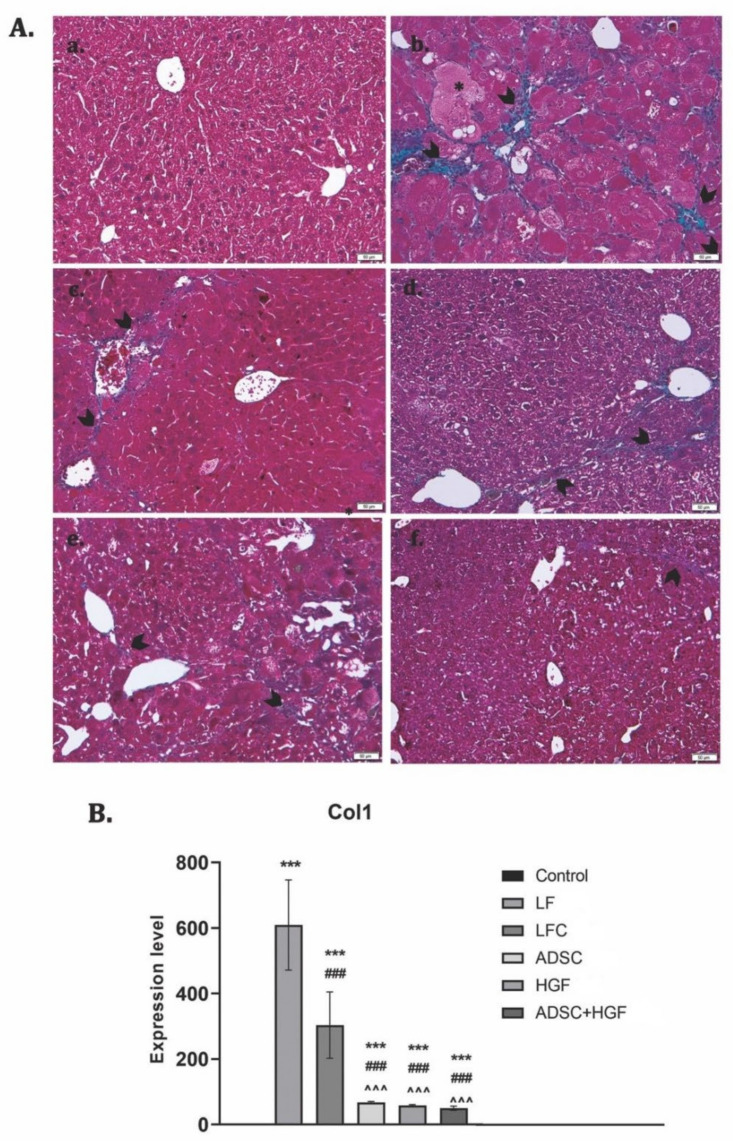
The effect induced by ADSC and HGF cotreatment on the reduction of collagen deposition in fibrotic livers of diabetic mice. (**A**) Gomori’s trichrome staining kit. (a) Control group: no significant collagen deposition; (b) LF group: significant collagen deposition with pseudo-lobular separation; (c) LFC group: aspect almost similar to the LF group; (d) ADSC and (e) HGF groups: less collagen deposition compared with the LF and LFC; (f) ADSC + HGF—few thin collagen septa; green collagen; scale bar: 50 μm. (**B**). RT-PCR analysis of collagen 1 (Col 1) gene levels. Legend: Control, LF—liver fibrosis in diabetic mice; LFC—self-recovery of liver fibrosis (positive control) in diabetic mice; ADSC—ADSC treatment of liver fibrosis in diabetic mice; HGF—HGF treatment of liver fibrosis in diabetic mice; ADSC + HGF—ADSC and HGF cotreatment of liver fibrosis in diabetic mice; *** *p* < 0.001 compared with the control; ### *p*< 0.001 compared with the LF group; ^^^ *p* < 0.01 compared with the LFC group.

**Figure 8 cells-11-03338-f008:**
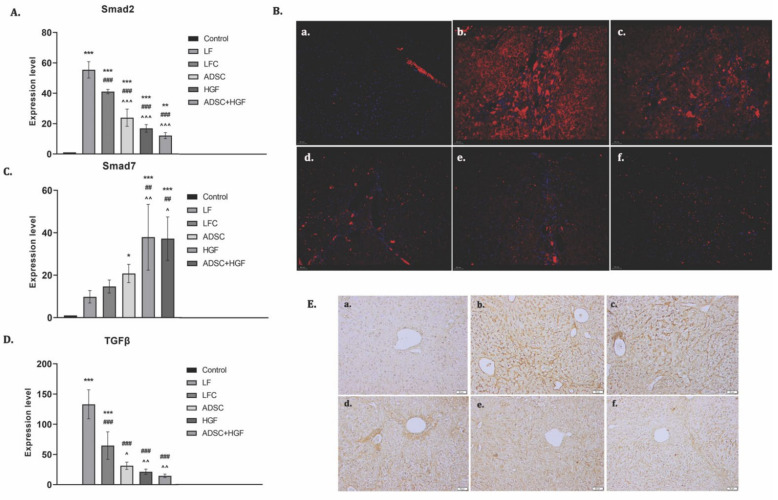
ADSC and HGF cotreatment inhibit profibrotic TGF-β1/Smad signaling. Legend: Control, LF group—liver fibrosis in diabetic mice; LFC group—self-recovery of liver fibrosis (positive control) in diabetic mice; ADSC group—ADSC treatment of liver fibrosis in diabetic mice; HGF group—HGF treatment of liver fibrosis in diabetic mice; ADSC + HGF group—ADSC and HGF cotreatment of liver fibrosis in diabetic mice (**A**). RT-PCR analysis of Smad2 gene level; *** *p* < 0.001 compared with control; ** *p* < 0.01 compared with control; ### *p* < 0.001 compared with the LF group; ^^^ *p* < 0.001 compared with the LFC group. (**B**). Immunofluorescence expression of Smad2 in experimental livers: (a) control, (b) LF, (c) LFC, (d) ADSC, (e) HGF, (f) ADSC + HGF. (**C**). RT-PCR analysis of the Smad7 gene level; *** *p* < 0.001 compared with control; * *p* < 0.05 compared with control; ## *p* < 0.01 compared with the LF group; ^ *p* < 0.05 compared with the LFC group. (**D**). RT-PCR analysis of the TGF-β1 gene level; *** *p* < 0.001 compared with control; ### *p* < 0.001 compared with the LF group**;** ^^ *p* < 0.01 compared with the LFC group; ^ *p* < 0.05 compared with the LFC group. (**E**) Immunohistochemical expression of TGF-β1 in experimental livers: (a) control, (b) LF, (c) LFC, (d) ADSC, € HGF, (f) ADSC + HGF. Scale bar: 50 μm.

**Figure 9 cells-11-03338-f009:**
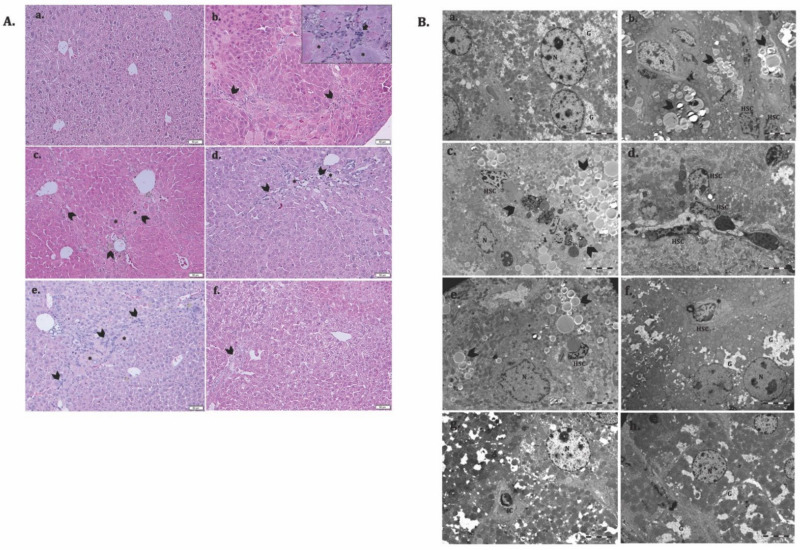
ADSC and HGF cotreatment improve the liver structure and ultrastructure of the fibrotic liver in diabetic mice. (**A**). Representative light microscopy micrographs of liver histology in H&E stain: (a) Control; (b) LF group—liver fibrosis in diabetic mice; (c) LFC group—self-recovery of liver fibrosis (positive control) in diabetic mice; (d) ADSC group—ADSC treatment of liver fibrosis in diabetic mice; (e) HGF group—HGF treatment of liver fibrosis in diabetic mice; (f) ADSC + HGF group—ADSC and HGF cotreatment of liver fibrosis in diabetic mice; scale bar: 50 μm; fibrosis (arrowhead), foam hepatocyte (*), inflammatory infiltrate (arrow); (**B**). Representative electron microscopy micrographs of the liver for the experimental groups; (a) control, (b–d) LF group, (e) LFC group, (f) ADSC group, (g) HGF group, (h) ADSC + HGF group. N-hepatocyte’s nuclei; HSC—hepatocyte stellate cells; lipids (arrowhead); collagen (*); glycogen (G).

**Table 1 cells-11-03338-t001:** Primer sequences for RT-PCR.

Target	Sense	Antisense
TGF-β1	5′ TTTGGAGCCTGGACACACAGTACi 3′	5′ TGTGTTGGTTGTAGAGGGCAAGGA 3′
α-SMA	5′ CCGACCGAATGCAGAAG GA 3′	5′ ACAGAGTATTTGCGCTCCGAA 3′
Smad 2	5′ GTTCCTGCCTTTGCTGAGAC 3′	5′ TCTCTTTGCCAGGAATGCTT 3′
Smad 3	5′ TGCTGGTGACTGGATAGCAG 3′	5′ CTCCTTGGAAGGTGCTGAAG 3′
Smad 7	5′ GCTCACGCACTCGGTGCTCA 3′	5′ CCAGGCTCCAGAAGAAGTTG 3′
Col I	5′ CAGCCGCTTCACCTACAGC 3′	5′ TTTTGTATTCAATCACTGTCTTGCC 3′
GAPDH	5′ CGACTTCAACAGCAACTCCCACTCTTCC-3′	5′ TGGGTGGTCCAGGGTTTCTTACTCCTT 3′

## Data Availability

Not applicable.

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
