# Peer review of "Adipose-Derived Stem Cells (ADSCs) Supplemented with Hepatocyte Growth Factor (HGF) Attenuate Hepatic Stellate Cell Activation and Liver Fibrosis by Inhibiting the TGF-β/Smad Signaling Pathway in Chemical-Induced Liver Fibrosis Associated with Diabetes"

_cells, 2022, doi:10.3390/cells11213338_

Round 1
Reviewer 1 Report (Previous Reviewer 1)
The article manuscript is the resubmitted version, which the editor rejects from IJMS (ijms-1834324).
A summary of the issues pointed out by the editor was as follows.
1. Abstract is too extended.
2. Western-blot analyses were recommended to show SMA and COLI proteins (Figures 1 and 2).
3. The necessity of ADSC+ HGF-DIA is unclear in Figures 5-8. Adding ADSC-HGF group is more suitable.
In the revised version, only issue 1 was clearly revised. Issues 2 and 3 still remain. Because the reviewer puts further consideration into the hand of the editors, the editor of IJMS rejected it. The main part of this resubmitted version is not improved from the reviewer’s view.
(Authors’ rebuttal against above comment 2)
At this moment, western blot can not be performed in our experimental setup due to the following reasons: (1) our in vitro experimental setup is a co-culture between ADSC and activated LX2 cells +/_ different treatment conditions, therefore cell lysis for western blot would be a mix of cells and protein expression irrelevant; (2) the quantity of cells needed to perform western blot is high; in these conditions, we can not simulate all the needed conditions, eg. treatment with HGF is not possible on such a surface and a high number of cells; (3) generally we consider that protein expression is a small part of the study just validating the other results and it does not represent the main purpose of our study, we have much more valuable data in the manuscript.
(The reviewer’s opinion in response to the rebuttal 2)
The reviewer understands the authors’ circumstances. However, the reviewer disagrees with the rebuttal.
(1) Please use mouse ADSC cells and monoclonal antibodies, which can only recognize human origins. How did the authors isolate RNAs from LX2 cells when the quantitative real-time PCR analyses to quantify SMA and COL1 mRNA expressions? In other words, how did the authors remove all human ADSC from the co-culture?
(2) In the reviewer’s experience, one plate of 35~60 mm dish is enough to prepare protein samples. Recombinant HGF is not too expensive.
(3) The reviewer thinks that all data should be scientifically comfortable. The immunofluorescence method is not suitable for quantifying expressed proteins.
(Authors’ rebuttal against above comment 3)
There is probably a confusion related to the in vivo experimental groups that we considered for our study and to the fact that ADSC are able to secrete HGF. HGF group is a target group since indeed we want to explore the hypothesis that HGF has a beneficial effect in liver fibrosis reversion. Apart from it, we have included an ADSC group meaning animals injected with ADSC considering the fact that ADSC secrete certain quantities of HGF, among others. Additionally, we have included ADSC+HGF group aiming to observe the cumulative effect of both HGF and ADSC secretome on fibrosis reversion. Therefore, considering this description, our experimental setup already includes the ADSC without HGF group, since our purpose was to observe ADSC effect alone or in combination to HGF. Basically we have included in the study ADSC treated group, HGF treated group and the combination ADSC+HGF group to see the cumulative beneficial effect of HGF and ADSC secretome.
(The reviewer’s opinion in response to the rebuttal 3)
The reviewer is afraid that he/she could not understand the authors’ logic. ADSCs secrete HGF. Therefore, the effect of ADSCs must be partially through HGF secreted from the ADSCs.
Finally, the authors’ rebuttals could not persuade the reviewer.
Author Response
Dear Reviewer 1,
The Authors of the manuscript entitled “Adipose-derived stem cells (ADSCs) supplemented with hepatocyte growth factor (HGF) attenuate hepatic stellate cell activation and liver fibrosis by inhibiting the TGF-β/Smad signaling pathway in chemical-induced liver fibrosis associated with diabetes”, submitted to Cells thank the referee for reviewing our manuscript. Please find attached the answers to all questions and suggestions.
Best regards,
The authors

Reviewer 2 Report (New Reviewer)
In the present manuscript, Gharbia et al., aimed to assess the anti-fibrotic effect of ADSCs and HGF therapy and to establish the cellular and molecular mechanisms, through in vitro and in vivo experiments.
Overall, the manuscript is well presented and the experimental design is structurally sound. However, there are some concerns that need to be addressed. The manuscript however is in need of significant language revision, the manuscript is full of grammatical and structural mistakes.
Page 1 Lines 51 & 52: In liver fibrosis, the cellular key player is represented by hepatic stellate cells (HSCs) which are activated and produce components of the extracellular matrix (ECM) that lead to the accumulation of scar tissue.
Ø Please further explain the mechanism underlying the HSCs role in liver fibrosis with references.
Page 1 line 77: “should patient’s donor stem cells would be used”
Line 81” angiogenesis regulator”
Ø Please check the sentence structure.
In the methodology section,
Ø LX2 cell line was provided by another lab. Kindly provide the identifying data of the cell line with reference.
In the discussion section,
Ø The authors mention that they evaluated the effect of ADSCs-secretome and hepatocyte growth factor (HGF) in vitro. It is not clear how was the secretome characterized exactly.
Author Response
Dear Reviewer 2
The Authors of the manuscript entitled “Adipose-derived stem cells (ADSCs) supplemented with hepatocyte growth factor (HGF) attenuate hepatic stellate cell activation and liver fibrosis by inhibiting the TGF-β/Smad signaling pathway in chemical-induced liver fibrosis associated with diabetes”, submitted to Cells thank the referee for reviewing our manuscript. We are deeply grateful for the advice and for the observations and comments that we addressed and feel that greatly increased the quality of our manuscript. Please find attached the answers to all questions and suggestions.
Best regards,
The authors

Reviewer 3 Report (New Reviewer)
The manuscript entitled Adipose-derived stem cells (ADSCs) supplemented with hepatocyte growth factor (HGF) attenuate hepatic stellate cell activation and liver fibrosis by inhibiting the TGF-β/Smad signaling pathway in chemical-induced liver fibrosis associated with diabetes focused on the ADSCs and HGF in HSC activation and liver fibrogenesis. The much appreciated work revealed the possibility of applying HGF-based ADSCs therapy in liver fibrosis associated with DM. Overall, the authors made contribution in the field, but there exists substantial room for improvement in the following areas:
Major concerns:
1. The authors aimed to imply the advantages of ADSCs supplemented with HGF in liver fibrosis intervention. However, in the results, the ADSC + HGF group didn’t show an evident effect compared with applying ADSC or HGF alone. According to the results, applying HGF alone and ADSC + HGF groups had the most obvious effect, which suggest that HGF might be the one that plays the most important role. Moreover, HGF administration could be much more convenient and economical when compared with the HGF-based ADSCs therapy.
2. HGF could be secreted by ADSCs. Could it be possible that ADSCs and HGF had a positive feedback loop which further enhance its protective role in liver fibrosis? This should be mentioned and discussed.
3. The results presented are confusing which reflected that the authors lack an overall recognition of the data. I propose to reconstruct the results in the following order:
1) Verification of fibrosis markers (Result 2.1 and 2.2);
2) As the important role of HSCs in liver fibrosis, I propose to present Result 2.5 and then 2.4;
3) As HSCs were activated by TGF-β1, the TGF-β/Smad signaling in fibrotic livers should be explored next (Result 2.6);
4) Finally, the overall effect of ADSCs and HGF co-treatment (Result 2.3).
Minor concerns:
1. The figure 1-3 exhibited one same result: IF of fibrosis markers (Result 2.1), which could be merged into one figure.
2. Results 2.4 should be ADSCs and HGF co-treatment suppress the production of the collagen in a liver fibrosis model of diabetic mice.
3. The authors used a carbon tetrachloride induced liver fibrosis model, a widely used fibrosis animal model. I would appreciate that if the authors discussed the possibility of applying HGF-based ADSCs therapy in liver fibrosis induced by various diseases, not limited to DM.
4. The authors should provide a list of abbreviations. And ones used could be presented in short (for example, HGF in line 418). Please check the manuscript carefully to ensure there do not exist such typo and grammar mistakes.
Author Response
Dear Reviewer 3,
The Authors of the manuscript entitled “Adipose-derived stem cells (ADSCs) supplemented with hepatocyte growth factor (HGF) attenuate hepatic stellate cell activation and liver fibrosis by inhibiting the TGF-β/Smad signaling pathway in chemical-induced liver fibrosis associated with diabetes”, submitted to Cells thank the referee for reviewing our manuscript. We are deeply grateful for the advice and for the observations and comments that we addressed and feel that greatly increased the quality of our manuscript. Please find attached the answers to all questions and suggestions.
Best regards,
The authors

Reviewer 4 Report (New Reviewer)
In this study, the authors demonstrated that ADSCs supplemented with HGF attenuate HSC activation and liver fibrosis in chemical-induced liver fibrosis associated with diabetes. This study found a potential treatment of liver fibrosis in diabetic patients, which is of great value. However, the following concerns should be addressed.
Major comments:
1. The reason why the authors combined the use of ADSC and HGF is not sufficient. Has this co-treatment therapy been used in previous studies? this should be addressed in the introduction part. In vitro, the effect of ADSC and HGF co-treatment was not measured, a ADSC and HGF co-treatment group is preferred to be added. In vivo, it seems that there is no significant difference in the collagen deposition and HSC activation in the ADSC+HGF-DIA group when compared to the ADSC-DIA group and HGF-DIA group, please further discuss the result.
2. In methods, liver fibrosis is commonly seen in type 2 diabetes, but in this study, the STZ injection, a type 1 diabetes model was applied. The reason why this model was used instead of type 2 diabetes models should be explained.
3. Result 2.1 and result 3.1 actually revealed the same conclusion that ADSCs and HGF inhibit HSC activation in vitro. These two paragraphs and figures are preferred to be combined into one.
4. In Result 2.1, it is inferred in the manuscript that “the secretome of ADSCs contained important anti-fibrosis factors that acted upon aHSCs. One of these factors could be HGF, as it reduced the expression of α-SMA in HSCs after treatment.”, this inference is not proved in this study as the authors did not measure the secretion of HGF by ADSCs and whether the inhibition effect of ADSCs on HSC activation would be reversed by reducing HGF secretion.
5. In result 2.3, the result of biochemical parameters is a bit confusing. The AST and ALT levels are higher in HGF-DIA and ADSC+HGF-DIA groups compared to the ADSC-DIA group and even LF-DIA group. And the LDH level is higher in the treatment group compared to LF-IDA and LFC-DIA groups. please explain it.
6. In result 2.5, it seems that the proliferation of HSCs is hard to recognize in the electron microscopy micrographs, can the authors explain this?
7. In result 2.6, TGF-β1/Smad signaling pathways are regulated by the phosphorylation of Smads proteins. Authors have conducted IF to examine the localization of Smad 2/3 in vitro, however, Smads were only measured at mRNA level in vivo. It would be better to add IF or western blot experiments in vivo. In addition, TGF-β and Smad inhibitors can be used to better confirm that TGF-β1/Smad signaling induces the treatment effect of ADSC and HGF.
Minor comments:
1. Line 182: “The treatments started on week 8…”, however, treatment was applied on week 7 in the following text, please check it and it would be better to provide a supplementary figure to introduce the design of in vivo experiment in order to help readers follow easily.
2. The subtitles are numbered wrongly in the result part. It should be 3.1, 3.2…
3. The language of the article needs to be improved, for example
Line 100: “hepatocyte growth factor (HGF)”, the abbreviation has been explained previously in line 95.
Line 106: “hepatic stellate cells (HSCs)” has also been explained before.
Line 341: “ADSCs+HG”, the spelling is wrong.
Line 390 and 391: “HGS”, the spelling is wrong.
Line 374: “a-SMA”, the spelling is wrong.
4. The labeling of α-SMA in the figure should be uniform. Alpha-SMA in figure1, while α-SMA in figure 4 and αSMA in figure 7.

Author Response
Dear Reviewer 4
The Authors of the manuscript entitled “Adipose-derived stem cells (ADSCs) supplemented with hepatocyte growth factor (HGF) attenuate hepatic stellate cell activation and liver fibrosis by inhibiting the TGF-β/Smad signaling pathway in chemical-induced liver fibrosis associated with diabetes”, submitted to Cells thank the referee for reviewing our manuscript. We are deeply grateful for the advice and for the observations and comments that we addressed and feel that greatly increased the quality of our manuscript. Please find attached the answers to all questions and suggestions.
Best regards,
The authors

Round 2
Reviewer 1 Report (Previous Reviewer 1)
The reviewer’s concerns were not improved in the revised version of the manuscript.
1. Even if they want to use the immunofluorescent (IF) method to quantify the protein amount, they need to show how they validate the quantifying method. Quantification of protein expression using IF is not easy to obtain reliable data. Did they validate their method? Please check the following paper.
Toki M et al., Proof of the quantitative potential of immunofluorescence by mass spectrometry. Lab Invest. 2017 Mar;97(3):329-334. doi: 10.1038/labinvest.2016.148. (PMID: 28092364)
In addition, please show how they distinguish the protein expression between human ADSC cells and LX2 cells.
2. The reviewer asks to add ADSC minus HGF-DIA group. It means the HGF-subtracted ADSC group does not ADSC without additional HGF (, which is simply the ADSC group). It can be archived by treating si/shRNA against HGF od ADSC or co-administration of anti-HGF antibody, which neutralizes HGF activity. ADSC must secrete HGF. Therefore, a synergistic effect of ADSC and HGF is not expected. Actually, the authors’ results show that the synergistic effect was not observed.
Author Response
Dear Reviewer1,
Thanks for taking the time to review this manuscript. However, the authors still keep the arguments they presented in the previous revisions.
Best regards,
The authors
Reviewer 3 Report (New Reviewer)
The revisions answered the major and minor concerns satisfactorily. The excellent work conducted by Sami Gharbia et al. reveals the potential of ADSC and HGF therapy in liver fibrosis associated with DM. I think the manuscript meets the standard of publication and agree to accept it.
Author Response
Dear Reviewer 3,
Thank you very much for the suggestions you gave us to improve the scientific quality of this article.
Best regards,
The authors
Reviewer 4 Report (New Reviewer)
Thanks to the authors for responding to my comments. The updates have addressed the majority of my concerns about the paper.
However, a few minor problems still exist:
1. The group names in each figure should be uniform. The group names in figure7(B) are not updated and still use the previous version.
2. Please rewrite the figure legend of Figure 8:
(1) The legend of figure 8(B) is missed while figure 8(C) is repeated.
(2) The legend of figure 8(D) should be placed after the legend of group names.
(3) Line 383: “LF-DIA” and” LFC-DIA” are still used.
(4) Line 384: “Legend: Legend:”, “Legend” is repeated.
(5) Line 388: “(e) HG, (f). ADSC+HG.”, “HG” should be “HGF”.

Author Response
Dear Reviewer 4,
Thank you for your observations and suggestions!
- The group names in each figure should be uniform. The group names in figure7(B) are not updated and still use the previous version.
- Please rewrite the figure legend of Figure 8:
(1) The legend of figure 8(B) is missed while figure 8(C) is repeated.
(2) The legend of figure 8(D) should be placed after the legend of group names.
(3) Line 383: “LF-DIA” and” LFC-DIA” are still used.
(4) Line 384: “Legend: Legend:”, “Legend” is repeated.
(5) Line 388: “(e) HG, (f). ADSC+HG.”, “HG” should be “HGF”.
We reviewed the text and made the changes you suggested!
Best regards,
The authors
This manuscript is a resubmission of an earlier submission. The following is a list of the peer review reports and author responses from that submission.
Round 1
Reviewer 1 Report
The authors investigated the anti-fibrotic effect of the adipose-derived stem cells (ADSCs) and hepatocyte growth factor (HGF) in TGF-beta1-activated hepatic stellate cells (HSCs) as an in vitro model and carbon tetrachloride-induced liver fibrosis in streptozotocin-induced type 1 diabetes model mice as an in vivo model. Co-culture of ADSCs or HGF treatment suppressed the active-HSC formation induced by TGF-beta1 (Figures 1-4). Furthermore, in the mice liver fibrosis model, ADSCs or HGF treatment suppressed liver fibrosis (Figures 5-7), at least in part by inhibiting TGF-beta/Smad 2 signaling pathway (Figure 8). Unfortunately, no description of the conclusion of their research either in the abstract, the discussion, or the conclusion section exists.
The results themselves seem to be appropriate, as already reported by others. However, unfortunately, the reviewer could not understand the purpose of the study after reading the manuscript. What is already known and still unknown? What are the authors’ research questions? To solve their research questions, what did they do? From the experimental results, how did they conclude? The authors need to show the above things in their manuscript clearly. In addition, the authors should solve the following issues before considering the publication of the manuscript in Biomedicines.
Title and Abstract
Although the authors used diabetic mouse models, they were streptozotocin-induced type 1 diabetes, and fibrosis was carbon tetra chloride-induced, not derived from a diabetic condition. Therefore, liver fibrosis used in the study was not associated with diabetes. In addition, cirrhosis and hepatocarcinoma are not directly related to the experimental results. The reviewer asks to describe the aim of the study, what they found new, and how they concluded against their purposes clearly in the abstract. The title should also be changed to reflect the abstract.
Introduction
1. Although the authors used streptozotocin-induced type I diabetes model mice, diabetes is not necessary to induce liver fibrosis with carbon tetrachloride. Also, ADSCs have been already reported to alleviate liver fibrosis induced by type 1 diabetes (PMID: 35365229), Echinococcus multilocularis (PMID: 35100287), NASH (PMID: 34926735), Cisplatin (PMID: 34572126), carbon tetrachloride (PMID: 33145022), type 2 diabetes model (PMID: 31747961). Therefore, please arrange the already known and unknown information to show the authors’ aim clearly. In addition, please describe what for did they perform in both in vitro and in vivo studies. At present, the reviewer does not feel the necessity of the in vitro experiments because the authors obtained the same results from both in vitro and in vivo experiments.
2. The reviewer does not understand why the authors used HGF in the study. Is it used as a positive control for alleviating liver fibrosis? Or did they want to show that HGF secreted by ADSCs is involved in improving liver fibrosis?
Results
1. Figures 1 and 2, is it possible to quantify the fluorescent intensities of SMA and Collagenase I signals? In addition, these experiments need an indication of the reference protein. Western blotting analyses may be more suitable for determining protein expression than immunofluorescence.
2. Figure 3, images of Hoechst are unclear, and the magnification of the photos seems low to show nuclear translocation. It is hard to show TGF-beta1-induced SMAD 2/3 nuclear translocation of HSCs.
3. Figures 1-4, LX2 + TGF-beta1 can be replaced by Activated LX2. Also, LX2 + ADSC and LX2 + HGF are replaced by Activated LX2 + ADSC and Activated LX2 + HGF, respectively. In this case, the term “Activated LX2” should be defined in the text clearly.
4. Figures 5-8, the reviewer could not understand the meanings of LFC-DIA group. Is the group necessary to lead to the authors’ conclusions in the study? The group might confuse the reading audience of the manuscript.
5. Figures 5-8, the reviewer could not understand the meanings of ADSC + HGF-DIA group. Did they expect synergistic effects by ADSC + HGF? If so, they should evaluate it. In the reviewer’s opinion, no need for the combination of ADSC and HGF.
Discussion
1. Lines 269-277, this sentence should be included in the introduction.
2. Line 278, if they showed in vitro data first in the world, they should reinforce the results and the usefulness more in the section.
3. Lines 280-300 and 325-333, as far as the reviewer read, essentially, the authors’ results of in vivo experiments are the same as the description. Even if they investigated the synergistic effect of ADSC + HGF, the impact was negative. Please clarify what is found new in the study.
Others
The conclusion should be added with an independent section.
Author Response
Dear Reviewer 1,
The Authors of the manuscript entitled “In vitro and in vivo studies supporting the therapeutic potential of adipose-derived stem cells (ADSCs) supplemented with hepatocyte growth factor (HGF) in liver fibrosis associated with diabetes”, submitted to IJMS thank the referee for reviewing our manuscript. We are deeply grateful for the advice and for the observations and comments that we addressed and feel that greatly increased the quality of our manuscript. Please find below the answers to all questions and suggestions.

Reviewer 2 Report
The manuscript Gharbia et al., titled: "In vitro and in vivo studies supporting the therapeutic potential of adipose-derived stem cells (ADSCs) supplemented with hepatocyte growth factor (HGF) in liver fibrosis associated with diabetes" is an interesting work investigating in in vitro and in vivo models how ADSC supplementation with HGF could constitute a therapeutic approach in liver fibrosis secondaty to Diabetes. This is an interesting work with potential clinical implications.
The reviewer would like to bring the following points to the attention of the authors:
1. The discussion may be interesting if also address the potential connection between diabetes progression and the microbiome.
2. How was the number of animals determined? Was there a power calculation for example? Please condised including this information in the manuscript accordingly.
3. How were the doses in all the experiments determined? (ie: What was the rationale for deciding on the doses used?). Please consider including this information in the manuscript appropriately.
4. Please provide the rationale for selecting the models used (cell lines and mouse strains).
5. In the introduction section in the last paragraph please clearly provide the principal hypothesis (Ho).
Author Response
Dear Reviewer 2,
We would like to thank you for the thorough reading of this manuscript and for your thoughtful comments and recommendations, which are of great help in revising the manuscript. We are also grateful for the positive feedback that we have received.
We agree with all of your comments and have revised our paper accordingly. Please find below our answers to your suggestions

Round 2
Reviewer 1 Report
The authors improved the manuscript in response to the reviewer’s comment. The reviewer now could understand the purpose and the conclusion of the study.
However, the following queries still remain for consideration for publication in IJMS.
1. Abstract is too extended. The length of the abstract should be a total of about 200 words maximum. Please make the abstract concise.
2. Figures 1 and 2, the reviewer understands that the expression of SMA and COLI proteins was obviously enhanced by TGFbeta-induced activation of satellite cells. These are confirmed by qPCR shown in Figure 4. However, in the reviewer’s opinion, quantified data in B do not sound scientific. Statistical comparison with multiple samples should be required for the quantification. In addition, these experiments are not for comparing the cellular distribution of the proteins. Therefore, the reviewer still recommends performing western blotting analyses with a reference protein, such as GAPDH, beta-actin, etc... Because these experiments do not use animals, it does not need approval from the ethical committee and can be done in a month.
3. Figures 5-8, the authors mentioned in the response to the reviewer that ADSC + HGF-DIA group was added to highlight that supplementation of the growth factor (HGF) enhances the antifibrotic effects and supports their hypothesis that HGF is the main responsible for the therapeutic effect (by ADSC). If so, ADSC – (minus) HGF-DIA group should be added instead of the ADSC + HGF-DIA group, shouldn’t it? The reviewer still could not understand the necessity of the ADSC + HGF-DIA group.
Author Response
To Reviewer 1,
Thank you for your valuable opinions and review on our paper. We appreciate your pieces of advice and your suggestions, please find below our answers:
- Indeed, the abstract was too extended and not precise enough for what we wanted to highlight in our study. Therefore, we have now shortened it and made it more concise, please find below the modified form of the abstract: „Liver fibrosis can develop on the background of hyperglycemia in diabetes mellitus. However, the xenobiotic-related factors may accelerate diabetes-associated liver fibrosis. In this study, we aimed to assess the antifibrotic effect of ADSC cell therapy in chemically-induced liver fibrosis in pre-existing diabetes conditions and to validate the hypothesis according to which HGF secreted by these stem cells can be the main responsible for this effect. In vitro, TGF-β1-activated hepatic stellate cells (HSCs) were co-cultured with ADSCs or HGF, and the expression of several fibrosis markers was investigated. The anti-fibrotic effect of the ADSCs, HGF and ADSCs supplemented with HGF, was further assessed in vivo on diabetic mice with liver fibrosis-experimentally induced. In vitro results showed the inhibition of HSCs proliferation and decrease of fibrogenesis markers. Co-administration of ADSCs and HGF on diabetic mice with liver fibrosis, enhanced anti-fibrotic effects confirmed by downregulation of Col I, α-SMA, TGF-β1 and Smad2, while Smad7 was upregulated. Moreover, stem cell therapy supplemented with HGF considerably attenuated inflammation and microvesicular steatosis, decreased collagen deposits and alleviates liver fibrosis. In conclusion, the HGF-based ADSCs therapy might be of interest for the treatment of liver fibrosis to diabetic patients, consecutive aggression exerts by different environmental factors.”
- There is probably a confusion related to the in vivo experimental groups that we considered for our study and to the fact that ADSC are able to secrete HGF. HGF group is a target group since indeed we want to explore the hypothesis that HGF has a beneficial effect in liver fibrosis reversion. Apart from it, we have included an ADSC group meaning animals injected with ADSC considering the fact that ADSC secrete certain quantities of HGF, among others. Additionally, we have included ADSC+HGF group aiming to observe the cumulative effect of both HGF and ADSC secretome on fibrosis reversion. Therefore, considering this description, our experimental setup already includes the ADSC without HGF group, since our purpose was to observe ADSC effect alone or in combination to HGF. Basically we have included in the study ADSC treated group, HGF treated group and the combination ADSC+HGF group to see the cumulative beneficial effect of HGF and ADSC secretome.
- Regarding the protein expression of SMA and Col-1, the Authors showed in the manuscript fibrotic markers gene expression accompanied by protein expression performed by immunostaining and confocal microscopy imaging. To fulfill the quantification of protein expression, confocal images were analyzed in ImageJ and statistics was performed comparatively in order to obtain relevant differences in protein expression between the studied conditions. We mention that at least 10 fields of view were quantified for each condition and that the results were statistically compared, as required. The Authors find the data relevant and completely scientific, as opposed to Reviewer’s opinion and numerous publications consider these data valid as protein expression. At this moment, western blot can not be performed in our experimental setup due to the following reasons: (1) our in vitro experimental setup is a co-culture between ADSC and activated LX2 cells +/_ different treatment conditions, therefore cell lysis for western blot would be a mix of cells and protein expression irrelevant; (2) the quantity of cells needed to perform western blot is high; in these conditions, we can not simulate all the needed conditions, eg. treatment with HGF is not possible on such a surface and a high number of cells; (3) generally we consider that protein expression is a small part of the study just validating the other results and it does not represent the main purpose of our study, we have much more valuable data in the manuscript. For all these reasons, the Authors respectfully ask the Reviewer to understand that we cannot perform the requested western blot analyzes due to technical inconsistencies. In future studies, we will optimize the experimental setup in such a manner to be able to perform western blot studies.

Reviewer 2 Report
The authors have reasonably addressed reviewer's comments. Proofreading is suggested.
Author Response
To Reviewer 2,
Thank you for your valuable opinions and review of our paper. We appreciate your pieces of advice and your suggestions.
Best regards,
The authors